



# Characterisation of Dansgaard-Oeschger events in palaeoclimate time series using the Matrix Profile

Susana Barbosa[1], Maria Eduarda Silva[1,2], and Denis-Didier Rousseau[3, 4, 5]

[1]INESC TEC, Porto, Portugal
[2]University of Porto, School of Economics and Management,Porto, Portugal
[3]Geosciences Montpellier, University of Montpellier, CNRS, France
[4]Institute of Physics, Silesian University of Technology, Gliwice, Poland
[5]Lamont-Doherty Earth Observatory, Columbia University, Palisades NY, USA

**Correspondence:** Susana Barbosa (susana.a.barbosa@inesctec.pt)

**Abstract.**

Palaeoclimate time series, reflecting the state of Earth's climate in the distant past, display occasionally very large and rapid shifts, evidencing abrupt climate variability. The identification and characterisation of these abrupt transitions in palaeoclimate records is of particular interest as it allows the understanding of millennial climate variability and the identification of potential tipping points in the context of current climate change. Methods that are able to characterise these events in an objective and automatic way, in a single time series or across two proxy records, are therefore of particular interest. In our study the matrix profile approach is used to describe Dansgaard-Oeschger (DO) events, abrupt warmings detected in Greenland ice core, and Northern Hemisphere marine and continental records. The results indicate that canonical events DO-19 and DO-20, occurring at around 72 and 76 ka, are the most similar events over the past 110,000 years. These transitions are characterised by matching transitions corresponding to events DO-1, DO-8 and DO-12. These transitions are abrupt, resulting in a rapid shift to warmer conditions, followed by a gradual return to cold conditions. The joint analysis of the $\delta^{18}$O and Ca$^{2+}$ time series indicates that the transition corresponding to the DO-19 event is the most similar event across the two time series.

## 1 Introduction

Palaeoclimate time series reflect the Earth's climate in the distant past based mainly on proxies derived from records such as sediments, ice cores or speleothems. One of the most extensively studied palaeoclimate time series is that of oxygen isotope ratios ($\delta^{18}$O) retrieved from Greenland ice cores in the context of the North Greenland Ice Core Project NGRIP (Andersen et al., 2004; Rasmussen et al., 2014). The concentration of $\delta^{18}$O, as measured in the ice, serves an indirect proxy for the air temperature at the ice core location.

The ice cores retrieved from the Greenland ice sheet revealed the occurrence of rapid warming events, which occurred over a few decades. Such abrupt transitions are designated as Dansgaard-Oeschger (DO) events, during which climate conditions alternated between full glacial (so-called stadial) and relatively mild (interstadial) conditions (Dansgaard et al., 1982; Johnsen et al., 1992; Rasmussen et al., 2014). These abrupt transitions, on average, are approximatively 12 °C, with a range of 5 °C





to 16 °C (Kindler et al., 2014). They exhibit a distinctive sawtooth shape, with rapid (decadal) increases in $\delta^{18}$O from GS (Greenland Stadial) to GI (Greenland interstadial) conditions, followed by slow relaxations back to GS conditions (centuries

or millennia). However, it should be noted that not all DO events have the same shape or duration (Rousseau et al., 2017; Lohmann, 2019).

DO events are particularly observable in the Greenland ice-core records, but similar transitions were identified in diverse palaeoclimate records (e.g. Rousseau et al. (2017); Corrick et al. (2020); Bagniewski et al. (2023); Held et al. (2024)) and thus DO events have been used for "wiggle-matching" of records with no accurate dating information (e.g Peterson et al. (2000);

Lauterbach et al. (2011)). The objective identification of DO events, in other than by visual inspection, is then of critical importance.

The original identification of DO events was conducted by visual inspection of the NGRIP $\delta^{18}$O time series. These "canonical DOs" (Rousseau et al., 2022) and further transitions were also identified visually by Rasmussen et al. (2014), who resorted not only to the $\delta^{18}$O record but also to the concurrent $Ca^{2+}$ proxy record. An algorithm was developed by Lohmann and

Ditlevsen (2019) for the characterisation of DO events based on their sawtooth shape. However, the approach does not identify the events themselves. Rather it relies on the previous visual identification by Rasmussen et al. (2014). A method based on the Kolmogorov–Smirnov (KS) test was developed by Bagniewski et al. (2021), which allows the identification of the canonical DO events in addition to other transitions that were previously unidentified by visual inspection. Although the method allows for the detection of individual jumps in the records, either towards cold or warm, or dry or wet conditions, it does not provide

any information on their magnitude. Recurrence plots (e.g. Marwan et al. (2007)) and measures of recurrence quantification analysis, such as the recurrence rate (Marwan et al., 2013) allow on the other hand for the identification of the dominant changes in a record's characteristic time scale (Bagniewski et al., 2021; Rousseau et al., 2023a, b).

Given that DO events can be considered as a recurring pattern in a palaeoclimate time series, algorithmic methods for the extraction of similar patterns from a time series can be applied to characterise these particular patterns. In this study the Matrix

Profile approach (Yeh et al., 2016) is employed to describe DO patterns in the Greenland ice core records. The methodology and data are described in sections 2 and 3, respectively. The results of the matrix profile approach applied to palaeoclimate time series are presented in section 4. Methodological and interpretative constraints are discussed in section 5, and concluding remarks are given in section 6.

## 2   Method

The matrix profile approach is described in detail in Yeh et al. (2016, 2018) and references therein. This overview provides only a brief summary of the methodology. Readers are referred to the original references for further details.

For a real-valued time series $T = \{t_1, ..., t_n\}$ of length $n$, a sub-sequence $T_{i,m}$ of length $m$ is a continuous subset of the values of $T$: $T_{i,m} = t_i, t_{i+1}, ..., t_{i+m-1}, 1 \leq i \leq n - m + 1$. The matrix profile is an ordered vector that stores the Euclidean distance from a sub-sequence $T_{i,m}$ to its closest (lowest distance) sub-sequence. The distance is measured using the Euclidean

distance between z-normalised sub-sequences with a mean of 0 and a standard deviation of 1 (Agrawal et al., 1993). The





matrix profile is defined as the distance between a sub-sequence and its closest sub-sequence, regardless of its location. That locations is stored in the profile index. The profile index is a companion vector taht stores the position of the nearest neighbour of each sub-sequence. The matrix profile and the matrix profile index are two data structures that annotate a time series with the distance and location, respectively, of the nearest neighbours of all its sub-sequences (in itself or in another time series).

The matrix profile enables the identification of motifs in the time series, corresponding to sub-sequences that are highly similar to each other. The (tying) lowest points in the matrix profile correspond to the locations of the optimal time series motif pair, i.e. the pair of sub-sequences that are most similar (Mueen et al., 2009): $\forall a,b,i,j \in [1,2,...,n-m+1],\ dist(T_{a,m},T_{b,m}) \leq dist(T_{i,m},T_{j,m})$, with $i \neq j, a \neq b, |i-j| \geq w, |a-b| \geq w, w > 0$, and $dist$ is the z-normalised Euclidean distance between the sub-sequences. The parameter $w$ imposes a constraint on the relative positions of the sub-sequences in the motif, ensuring a

gap between the sub-sequences of at least $w$ values. This exclusion zone is intended to exclude trivial motifs as described in Lin et al. (2002).The exclusion zone thus defined enables the avoidance of trivial matches of a sub-sequence with itself or of largely overlapping sub-sequences. For a top motif of the closest sub-sequences, which are distanced by $D_{1,min}$, neighbouring similar sub-sequences can be obtained by considering the sub-sequences within $D_{1,min} \times R$ distances, where $R$ is a small integer value (typically $\leq 2$). The definition of a first order motif can be generalized to that of a $k$-th order subsequent motif

(where $k$ is an integer number) by considering the following nearest pair of sub-sequences, after excluding all sub-sequences belonging to prior motifs.

For a single time series $T$ the matrix profile is a vector of the Euclidean distances between the closest sub-sequences, considering all possible sub-sequences of $T$ obtained by sliding a window of length $m$ across $T$. The concept can be extended to two time series $T_A$ and $T_B$ (which may have different lengths), with the join matrix profile containing the Euclidean distances

between every sub-sequence in $T_A$ and its closest sub-sequence in $T_B$. The join motifs reflect the most similar patterns between the two time series (Yeh et al., 2017, 2018).

The python code stumpy (Law, 2019) is employed to perform the matrix profile analysis. The R Statistical Software (version 4.1.3 , R Core Team (2022)) is employed for the visualisation of the matrix profile results.

## 3   Data

The analysis presented here applies to the 20-year resolution time series of $\delta^{18}O$ and calcium ion ($Ca^{2+}$) concentrations from the NGRIP ice core, on the GICC05modelext time scale (Rasmussen et al., 2014; Seierstad et al., 2014). The time series encompasses the interval from 107.6 to 10.2 ka b2k (before A.D. 2000), 4869 data points in total. While oxygen isotope ratio is a proxy for air temperature, calcium is a proxy of atmospheric dust (Fuhrer et al., 1999), reflecting changes in dust sources and transport pathways (Fischer et al., 2007). The time series are presented in Figure 1, with the $Ca^{2+}$ record represented on a

reverse logarithmic scale, in line with the approach outlined by Rasmussen et al. (2014). A small number of missing values in the $Ca^{2+}$ time series, $< 1\%$, were linearly interpolated.

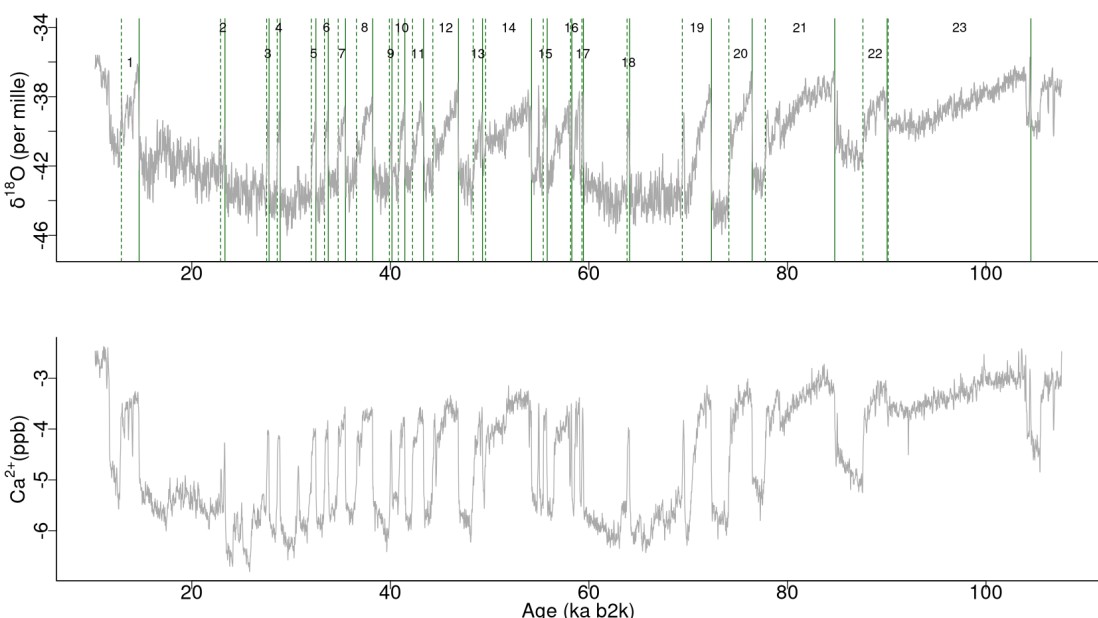

**Figure 1.** Time series of $\delta^{18}$O and $Ca^{2+}$ concentrations on the GICC05modelext time scale. Note that Ca2+ is in inverse logarithmic scale. The vertical solid and dashed lines represent, respectively, the start and end times of canonical DO events (Dansgaard et al., 1993; Rasmussen et al., 2014).

## 4  Results

The results of the matrix profile analysis of the palaeoclimate time series are presented initially in terms of the characterisation of DO events from a single time series, the $\delta^{18}$O Greenland record, in section 4.1, and the $Ca^{2+}$ record, in section 4.2. Subsequently, the joint analysis of DO events from multiple time series is documented in section 4.3 via both the $\delta^{18}$O and the $Ca^{2+}$ proxy records.

### 4.1  Matrix profile analysis of the $\delta^{18}$O record

The matrix profile of the $\delta^{18}$O time series is computed using the *stomp* (Scalable Time series Ordered-search Matrix Profile) algorithm (Zhu et al., 2016) with a sub-sequence length (window size $m$) of 2,500 years (125 data points). The exclusion zone, defined as a proportion of the sub-sequence length, $w/m$ is set as 1, that is equal to the size of the sub-sequence length, in order to exclude trivial matches (similarity of a sub-sequence with another sub-sequence with data values in common). For each $ith$ sub-sequence consisting in the time series values from $i$ to $i+m-1$, with $i=1,2,...n-m+1$, its z-normalised Euclidean distance to every other $j^{th}$ sub-sequence is computed. The matrix profile is defined as the smaller value from that set of distances, while the profile index records the location of the closest (non-trivial) sub-sequence, or in other words, the value of $j$. Table 1 provides an illustrative example of the matrix profile and corresponding profile index computed for the $\delta^{18}$O time series.





**Table 1.** Snippet of the matrix profile and profile index of the $\delta^{18}$O time series. Values corresponding to the global minimum of the matrix profile are represented in bold.

| Sub-sequence<br>($i$ (time, ka b2k)) | Matrix profile<br>(distance) | Profile index<br>($j$ (time, ka b2k)) |
| --- | --- | --- |
| ... | | |
| 3044 (71.12) | 2.37 | 3250 (75.24) |
| **3045 (71.14)** | **2.34** | **3251 (75.26)** |
| 3046 (71.16) | 2.38 | 3252 (75.28) |
| 3047 (71.18) | 2.50 | 3253 (75.30) |
| 3048 (71.20) | 2.59 | 3254 (75.32) |
| 3049 (71.22) | 2.60 | 3255 (75.34) |
| ... | | |

Figure 2 depicts the complete matrix profile for the $\delta^{18}$O time series, which represents the distance of each sub-sequence to its closest (non-trivial) sub-sequence. The global minimum value of the matrix profile is 2.34, occurring at 71.14 ka, as represented in Figure 2 by a dashed vertical line. Of all the sub-sequences in the $\delta^{18}$O time series, the one starting at 71.14 ka

exhibits the lowest distance to its closest sub-sequence, namely the one starting at 75.26 ka (Table 1). It should be noted that this sub-sequence does not necessarily correspond to the next minimum value of the matrix profile. Rather, it is the sub-sequence that is the closest to the one extracted from the global minimum, and not necessarily the sub-sequence with the second-lowest matrix profile distance.

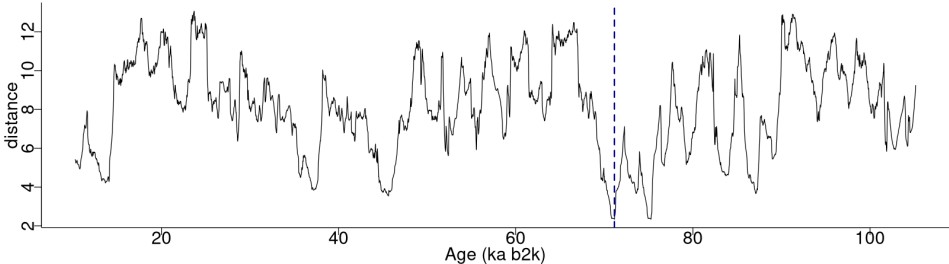

**Figure 2.** Matrix profile of the NGRIP $\delta^{18}$O time series using a window size of 2,500 years (125 data points). The dashed vertical line indicates the minimum value of the matrix profile corresponding to the top motif. Same age model than in Figure 1.

The top motif pair, representing the two closest sub-sequences in the time series, starting at 71.14 ka and 75.26 ka, respec-

tively, is displayed in Figure 3. This motif pair corresponds to the canonical DO events DO-19 and DO-20, which are the most similar to each other. While the sub-sequence starting at 75.26 ka is the closest to the top sub-sequence corresponding to the matrix profile minimum, at 71.14 ka, other sub-sequences, although not the closest, are still not far off. All the sub-sequences for which the distance to the top sub-sequence at 71.14 ka is within $R$ times the distance between the top motif pair can be

considered as neighbouring motifs. Table 2 presents the motifs obtained for the $\delta^{18}O$ time series using values of $R$ equal to
2 and 3. Setting a value of $R = 2$ enables the extraction of three neighbouring motifs to the top motif, while setting $R = 3$
allows for the additional extraction of five further neighbours. It should be noted that the distance represented in Table 2 is the
distance of the motif sub-sequence to the top motif, and not its matrix profile distance. An alternative approach to the use of
a fixed value of the parameter $R$ is to set a maximum distance between a neighbour motif and the main motif based on the
variability of the matrix profile. In this study, neighbour motifs are considered to be distanced by less than twice the global
standard deviation of the matrix profile (equal to $2 \times 2.38 = 4.76$), which in this case corresponds to the neighbours that would
be extracted by considering $R = 2$ (Fig. 3). This aspect of selecting neighbour sequences to the main motif is further discussed
in section 5. The last column of Table 2 indicates the canonical DO event that coincides with each motif. The seventh motif
(12.98 - 10.48 ka) does not correspond to a DO event but it does include the Younger Dryas cooling event (GS-1) and the
transition to the Holocene.

Figure 4 depicts the normalised sub-sequences corresponding to the motifs extracted from the $\delta^{18}O$ time series. The top motif
represents the patterns corresponding to canonical DOs 19 and 20. These aforementioned sub-sequences and its neighbour
motifs represent abrupt transitions to warmer conditions, preceded by an approximately stable stadial level and followed by a
slow decrease towards stadial conditions.

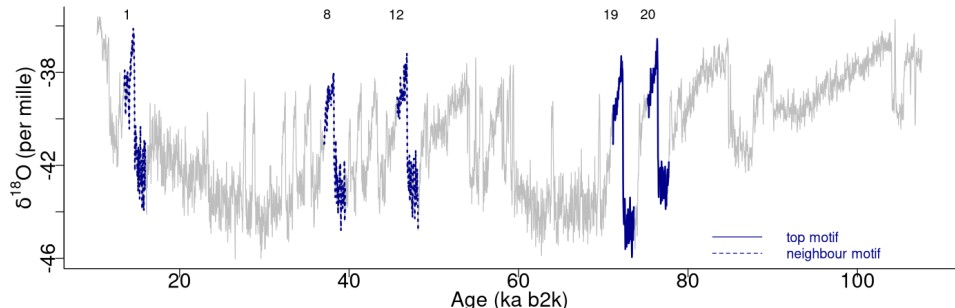

**Figure 3.** Top motifs for the NGRIP $\delta^{18}O$ time series and neighbour motifs. The horizontal top numbers indicate the canonical DO events.
Same age model than in Figure 1.

## 4.2  Matrix profile analysis of the $Ca^{2+}$ record

A comparable analysis is conducted for the $Ca^{2+}$ record employing the same methodology as that employed for the $\delta^{18}O$ time
series. Figure 5 depicts the matrix profile for the $Ca^{2+}$ record. While the overall pattern of the $Ca^{2+}$ series is similar to that
of the matrix profile of the $\delta^{18}O$ record in Figure 2, the peaks are sharper, due to the lower noise level of the $Ca^{2+}$ series.
Moreover, the minimum value of the matrix profile for the $Ca^{2+}$ series, indicated by the vertical dashed line in Figure 5, occurs
at a different time, at 37.14 ka, than for the $\delta^{18}O$. Consequently, Thus, the sub-sequence with the smallest distance to its closest
sub-sequence in the $Ca^{2+}$ record starts at 37.14 ka, corresponding to DO-8. From the profile index, the closest sub-sequence,
which constitutes the top motif pair for the $Ca^{2+}$ time series, starts at 45.78 ka, corresponding to DO-12. The aforementioned

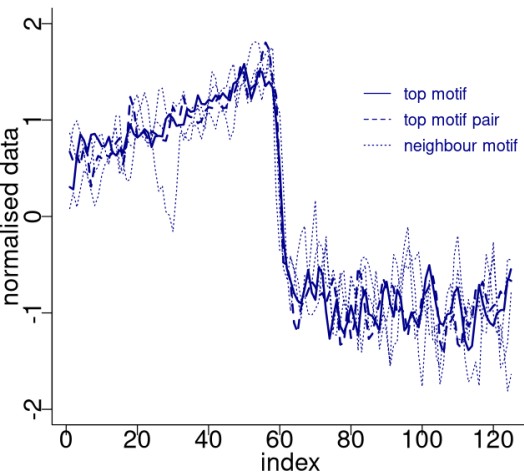

**Figure 4.** Normalised motifs for the NGRIP $\delta^{18}$O time series. The index is the order of the $\delta^{18}$O values in each sub-sequence of 125 values (2,500 years). The top motif pair corresponds to DO 19 and 20 and neighbour motif corresponds to DO 12, 8 and 1.

**Table 2.** Motifs extracted from the matrix profile of the $\delta^{18}$O time series using a window size of 2,500 years. The columns display: the value of the radius parameter $R$ for neighbouring sub-sequences, the type of motif, end and start time of the motif, the distance to the top motif, and the corresponding canonical DO event.

| R | Motif | Time interval (ka b2k) | | Distance | Event |
|---|---|---|---|---|---|
| | | End time | Start time | | |
| | 1- top | 71.14 | 73.64 | 0 | DO-19 |
| | 2 - top (pair) | 75.26 | 77.76 | 2.34 | DO-20 |
| 2, 3 | 3 - neighbour | 45.68 | 48.18 | 3.83 | DO-12 |
| | 4 - neighbour | 37.06 | 39.56 | 3.94 | DO-8 |
| | 5 - neighbour | 13.48 | 15.98 | 4.45 | DO-1 |
| | 6 - neighbour | 83.58 | 86.08 | 4.81 | DO-21 |
| | 7 - neighbour | 10.48 | 12.98 | 5.35 | - |
| 3 | 8 - neighbour | 88.86 | 91.36 | 6.43 | DO-22 |
| | 9 - neighbour | 102.84 | 105.34 | 6.50 | DO-23 |
| | 10 -neighbour | 52.94 | 55.44 | 6.90 | DO-14 |

top motif, and its neighbour motifs are summarised in Table 3. The criterion of selecting as neighbour motifs the sub-sequences distanced from the main motif by less than twice the matrix profile's standard deviation ($2 \times 2.50 = 5$), yields all neighbour motifs for a radius $R = 2$, as well as the first neighbour motif for a value of $R = 3$. This is illustrated in Figure 6.


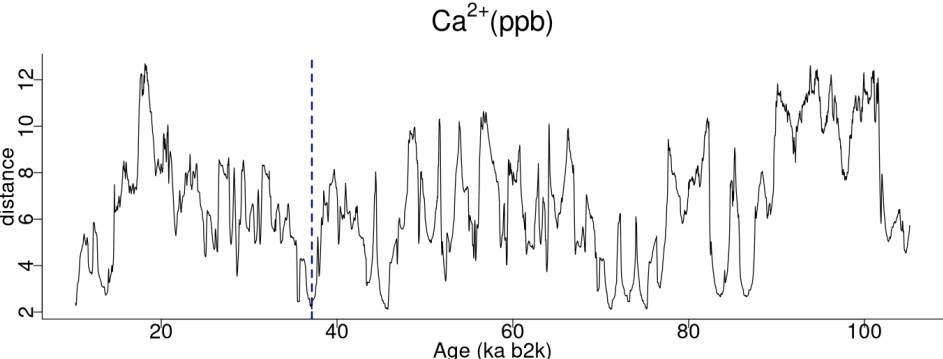

**Figure 5.** Matrix profile of NGRIP $Ca^{2+}$ record. Same conventions than in Figure 2.

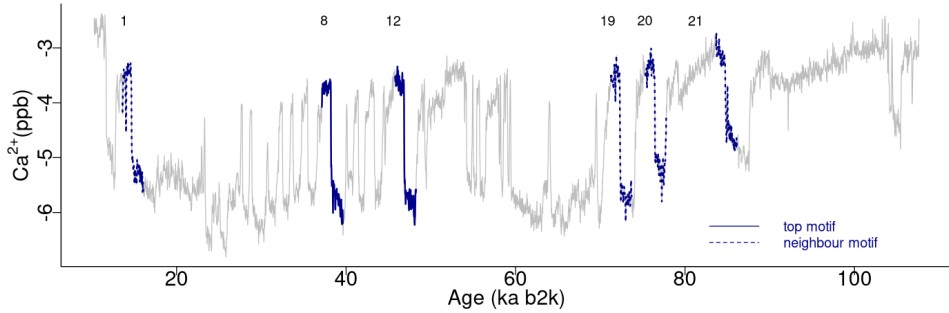

**Figure 6.** Top motifs for the NGRIP $Ca^{2+}$ record. Same conventions than in Figure 3.

Figure 7 displays the normalised motifs for the $Ca^{2+}$ time series. These motifs correspond to the same canonical DOs as the motifs of the $\delta^{18}O$ time series, with the addition of DO-21. However, the ordering of the motifs differs. In the case of the $Ca^{2+}$ record, considered in reverse logarithmic scale, the recurring patterns, constituted by the top motif and its neighbours motifs, represent abrupt decreases in the atmospheric dust concentration. These decreases are preceded by slightly decreasing stadial values on the plot and followed by an approximately stable interstadial level.

### 4.3 Join matrix profile analysis of the $\delta^{18}$ and $Ca^{2+}$ records

Figure 8 depicts the join matrix profile of the $\delta^{18}O$ and $Ca^{2+}$ time series, which is defined as the distance between each sub-sequence in the $\delta^{18}O$ record and its closest sub-sequence in the $Ca^{2+}$ record. In this join case, the necessity for an exclusion zone to avoid trivial matches is negated by the fact that the sub-sequences originate from different time series. The join matrix profile (Figure 8) differs from the previously defined individual profiles but exhibits similarities to the matrix profile for the

single $\delta^{18}O$ time series (Figure 2). This is not unexpected given that the temporal variability of the $Ca^{2+}$ series closely follows that of the $\delta^{18}O$ series. For illustrative purposes, a snippet of the join matrix profile and profile index is presented in Table 4.


**Table 3.** Motifs extracted from the matrix profile of the $Ca^{2+}$ record. Same conventions than in Table 2.

| R | Motif | Time interval (ka b2k) | | Distance | Event |
|---|---|---|---|---|---|
| | | End time | Start time | | |
| | 1- top | 37.14 | 39.64 | 0 | DO-8 |
| | 2 - top (pair) | 45.78 | 48.28 | 2.12 | DO-12 |
| 2, 3 | 3 - neighbour | 71.24 | 73.74 | 2.46 | DO-19 |
| | 4 - neighbour | 13.6 | 16.1 | 2.86 | DO-1 |
| | 5 - neighbour | 83.66 | 86.16 | 2.88 | DO-21 |
| | 6 - neighbour | 75.3 | 77.8 | 3.65 | DO-20 |
| | 7 - neighbour | 102.96 | 105.46 | 5.17 | DO-23 |
| 3 | 8 - neighbour | 10.56 | 13.06 | 5.32 | - |
| | 9 - neighbour | 52.92 | 55.42 | 6.06 | DO-14 |
| | 10 -neighbour | 88.92 | 91.42 | 6.32 | DO-22 |

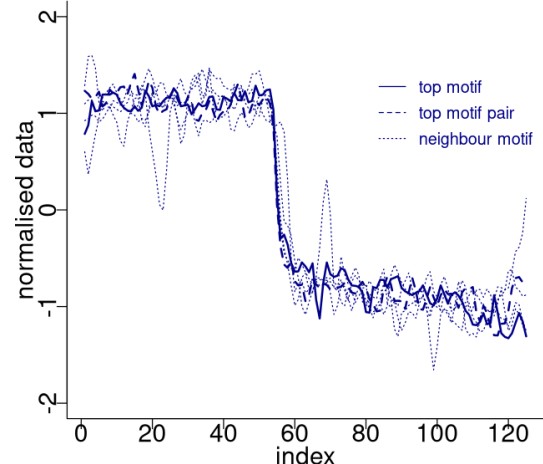

**Figure 7.** Normalised motifs for the NGRIP $Ca^{2+}$ record. Same conventions than in Figure 4, but top motif pair corresponds to DO8 and 12 and neighbour to DO19,1,21 and 20. The original data is in reverse logarithmic scale.

The minimum value of the join matrix profile indicates the location of the top motif, that is, the sub-sequence in the $\delta^{18}O$ time series which is closest (distance-wise) to a sub-sequence in the $Ca^{2+}$ time series (whatever it is). The global minimum of the join matrix profile occurs at 71.28 ka, which is in close proximity to the previously defined minimum for $\delta^{18}O$ at 71.14 ka b2k. Table 4 indicates that the sub-sequence in the $\delta^{18}O$ time series starting at 71.28 ka exhibits the greatest similarity to sub-sequence in the $Ca^{2+}$ record also starting at 71.28 ka. The most analogous sub-sequences across the $\delta^{18}O$ and $Ca^{2+}$ records are presented in Figure 9, still with $Ca^{2+}$ displayed in reverse logarithmic scale. In this particular case, the location



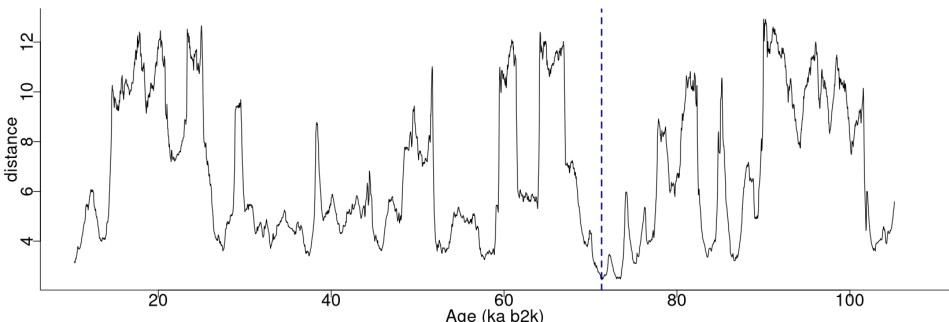

**Figure 8.** Join matrix profile for the $\delta^{18}$O and $Ca^{2+}$ time series using a window size of 2,500 years (125 data points). The dashed vertical line indicates the minimum value of the matrix profile corresponding to the top motif (sub-sequence of the $\delta^{18}$O time series closest to a sub-sequence in the $Ca^{2+}$ time series). Same age model than in Figure 1.

**Table 4.** Snippet of the join matrix profile and profile index of the $\delta^{18}$O and $Ca^{2+}$ time series.

| Sub-sequence ($i$ (time, ka b2k)) | Join matrix profile (distance) | Profile index ($j$ (time, ka b2k)) |
|---|---|---|
| ... | | |
| 3049 (71.22) | 2.50 | 3049 (71.22) |
| 3050 (71.24) | 2.49 | 3050 (71.24) |
| 3051 (71.26) | 2.46 | 3051 (71.26) |
| **3052 (71.28)** | **2.44** | **3052 (71.28)** |
| 3053 (71.30) | 2.45 | 3053 (71.30) |
| 3054 (71.32) | 2.46 | 3054 (71.32) |
| ... | | |

(time) of the top motif is the same for the two records, and thus normalised motifs are represented on the temporal scale rather than plotted as a function of the index value from 1 to $m = 125$, as in Figures 4 and 7). This motif is identical to the previously identified motif in the individual analysis of the $\delta^{18}$O time series (section 4.1). This transition, occurring approximately 70,000 years ago (canonical DO-19), exhibits the most analogous patterns of warming/cooling across the $\delta^{18}$O and less dusty/ dustier $Ca^{2+}$ records. Additionally, it also bears the closest resemblance in shape to the transition in the $\delta^{18}$O record immediately after, approximately 73,000 years ago.

## 5 Discussion

While the matrix profile is an algorithmic approach that enables the extraction of recurring patterns in a time series, its results are dependent on parameters that are prescribed empirically, often through a trial-and-error process, and are dependent on the

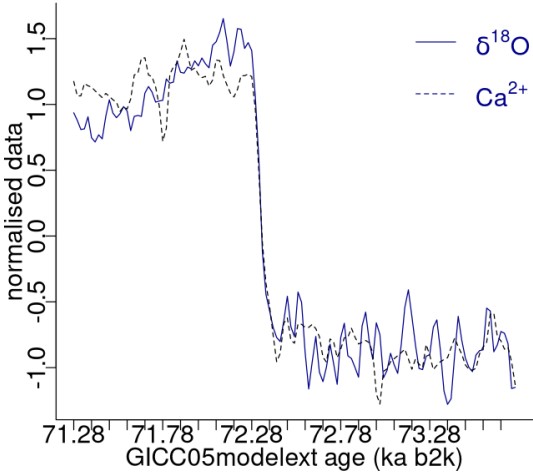

**Figure 9.** Normalised top motif across the NGRIP $\delta^{18}$O and Ca$^{2+}$ time series. Same age model than in Figure 1.

specific application and purpose of the analysis. These constraints are discussed in section 5.1, while section 5.2 discusses the extraction of the most similar pattern across two time series.

## 5.1 Matrix profile parameters

The matrix profile is dependent on a single parameter, the sub-sequence length $m$. Consequently, it is a method that can be readily applied to any time series, as it only requires the appropriate specification of a window size (see section 4.1). However, the matrix profile results are highly contingent upon the value selected for $m$. There is no universal formula or rigorous criteria for selecting the window size, as it depends on the objective of the analysis and the type of motif being investigated. The sub-sequence length is typically determined by considering the length of the patterns of interest in the time series being analysed. In

the absence of prior knowledge regarding the length of the motif of interest in a time series, an extension of the matrix profile, which computes nearest neighbour information for all sub-sequences of all lengths, can be considered (Madrid et al., 2019).

In our study, a window size of 2,500 years was selected as adequate for the extraction of motifs with duration typical of DO events. The matrix profile obtained with this window size is compared in Figure 10 with the matrix profile for window sizes of 3,000 and 3,500 years. The matrix profile values increase with increasing value of $m$, yet the results remain remarkably

consistent across this range of window sizes, exhibiting a similar overall pattern and exhibiting the lowest distance at around 70 ka. The highest vales of the matrix profile (at around 90 ka, 66 ka and 26 ka) reflect the flatter parts of the $\delta^{18}$O time series, which lack discernible features and thus comprise more disparate sub-sequences. The motifs obtained for the different window sizes are presented in Figure 11, which provides further evidence of the robustness of the recurring patterns extracted in the considered window range.

The top motif is derived directly from the lowest values of the matrix profile, and therefore depends only on the specified sub-sequence length. However, the extraction of neighbouring sequences to the top motif and of other motifs depends not only


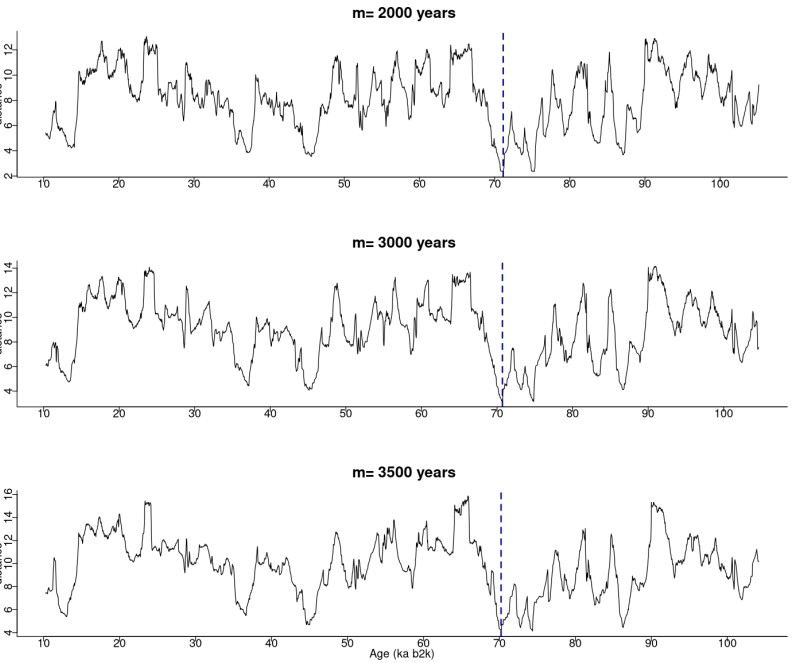

**Figure 10.** Matrix profile of the NGRIP $\delta^{18}$O time series for different window sizes. The dashed vertical line indicates the minimum value of the matrix profile. Same age model than in Figure 1.

on the window size but also on the tolerance (distance-wise) to which a sub-sequence is considered to match a pattern, given by the value of parameter $R$. Once more, there are no specific criteria for specifying $R$, other than that it should be a small integer value. The larger the value of $R$, the greater the distance between sub-sequences considered to match the motif, and therefore

the lower the similarity between potential matches. The use of smaller values of $R$ reduces the potentiality for considering as matching sub-sequences that are dissimilar. This is demonstrated in Figure 12, which depicts the normalised motifs for the NGRIP $\delta^{18}$O time series using R=2, as well as the supplementary motifs obtained with R=3. To facilitate visualisation, the first two neighbour motifs for R=3 are plotted separately from the subsequent neighbour motifs (see Table 2). The discrepancy between the top motif pattern and neighbour motifs assigned increases with the higher-order motifs indicating that a value of

$R = 3$ is not optimal in this case. Notwithstanding, the neighbour motifs 6 and 7, derived from the use of R=3, remain in close proximity to the primary motif. An alternative approach to enhance the flexibility of the extraction of neighbour motifs is to consider a maximum distance not calculated from a radius $R$, but rather from a metric computed from the overall matrix profile variability. In this study we adopt this approach, whereby neighbour motifs are selected as the patterns that are distanced from the top motif by less than twice the standard deviation of the matrix profile. For the $\delta^{18}$O time series, the motifs obtained using

this criterion are identical to those obtained by setting $R = 2$. However, this approach yields same neighbour motifs for the $Ca^{2+}$ time series as for $R = 2$, in addition to the first neighbour motif corresponding to $R = 3$.


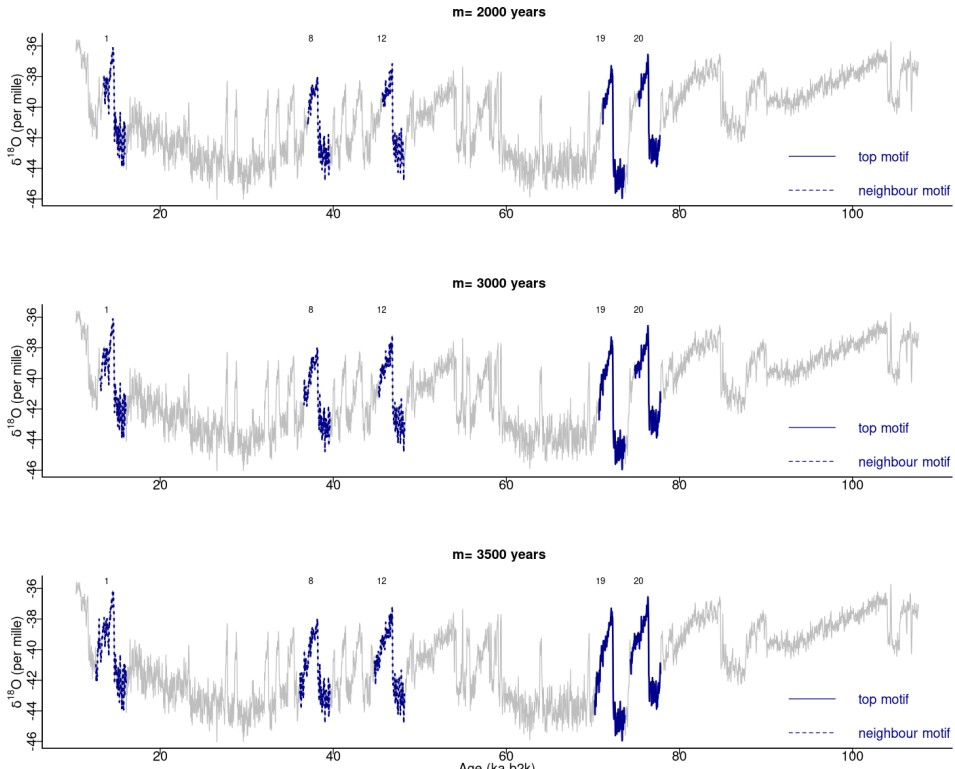

**Figure 11.** Motifs extracted from the matrix profile of the NGRIP $\delta^{18}$O time series for different window sizes. Same age model than in Figure 1.

The identification of higher-order motifs, other than the top (global) motif corresponding to the matrix profile minimum), is more challenging, due to the dependence on the constraints that must be be set in terms of the maximum distance for which sub-sequences are taken as matching. As illustrated in the preceding section, varying the radius parameter yields both matching and dissimilar patterns. Therefore, we have adopted a more flexible criterion that that of the radius parameter $R$ in order to ensure appropriately constrained results and interpretable motifs. Subsequent motifs depend on the specified distance constraints. To circumvent this limitation, this study focuses on the extraction of the top motif from the palaeoclimate records.

The comparison of sub-sequences similarity is performed here based on the Euclidean distance metric. An alternative would be to consider instead dynamic time warping (DTW) as a more robust distance measure for time series (Keogh and Ratanama-hatana, 2005). However, empirical comparisons showed that the Euclidean distance is competitive with or superior to more complex measures (Ding et al., 2008). Therefore, the Euclidean distance between z-normalised sub-sequences is employed as the similarity measure in this study.

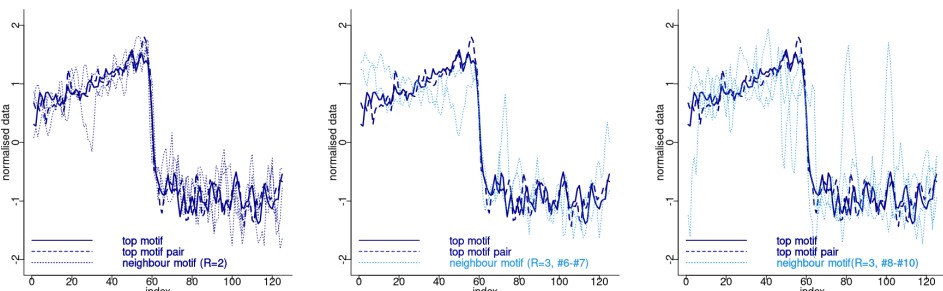

**Figure 12.** Normalised motifs for the NGRIP $\delta^{18}$O time series: top motif and neighbour motifs 3 to 5 (see Table 2 corresponding to R=2 (left), top motif and neighbour motifs 6 and 7 (middle), top motif and neighbour motifs 8 to 10 (right). Same convention than in Figure 4.

## 5.2 Extraction of dominant motif across two time series

The most analogous pattern (top motif) across two time series is not required to occur at the same time in the two series. This
fact is illustrated by considering the same $\delta^{18}$O and $Ca^{2+}$ proxy records, but artificially changing the $Ca^{2+}$ time series in two distinct ways. The first case introduces an artificial shift in time of 500 data points (10 ka) by adding to the beginning of the record 500 points with the same value as the mean of the $Ca^{2+}$ time series. The second case removes the first 500 data points of the $Ca^{2+}$ series, thus obtaining a shorter record, of different length than the $\delta^{18}$O series. The time series are displayed in Figure 13, and the join matrix profile between the $\delta^{18}$O series and these versions of the $Ca^{2+}$ series is presented in Figure 14. The
outcomes are largely analogous, with only minor discrepancies in the join matrix profile for the different versions of the $Ca^{2+}$ time series. In the case of the shifted version of the $Ca^{2+}$ series (Figure 14b), the difference to this version and the original is observed in the highest values of the join matrix profile, which are flatter. The matrix profile peaks correspond to parts of the series with no discernible temporal structure, typically a featureless noise level. In such cases the closest sub-sequences are the stable level values (equal to the mean of the time series) that have been introduced in the beginning of the record. In the case of
the trimmed version of the $Ca^{2+}$ series (Figure 14c), it should be noted that the join matrix profile has the same length, despite the reduced length of the $Ca^{2+}$ series. This is because the join matrix profile contains the distance of every sub-sequence in the $\delta^{18}$O series to the trimmed $Ca^{2+}$ series (of smaller size). Consequently, the number of sub-sequences (in the $\delta^{18}$O series) remains constant, yet the distances are calculated between each of these sub-sequences and a smaller number of sub-sequences in the $Ca^{2+}$ series. The principal distinction between the join matrix profile values in this particular case is observed at the
beginning of the record, as the initial sub-sequences of the $\delta^{18}$O series are the closest to the initial sub-sequences in the $Ca^{2+}$ series, which no longer exist, having been matched to other sub-sequences in the $Ca^{2+}$ time series.

For the original and the two versions of the $Ca^{2+}$ under consideration, the minimum value of the join matrix profile occurs at the same time corresponding to the top motif previously identified in the $\delta^{18}$O record, at around 71 ka. Tables 5 and 6 present excerpts of the join matrix profile and profile index for the shifted and trimmed $Ca^{2+}$ time series, respectively. In the former
case, the minimum value of the matrix profile occurs at the same index value of the $\delta^{18}$O time series as previously defined. However, the closest sub-sequence in the $Ca^{2+}$ record is no longer coincidental, although it is correctly identified at 10 ka

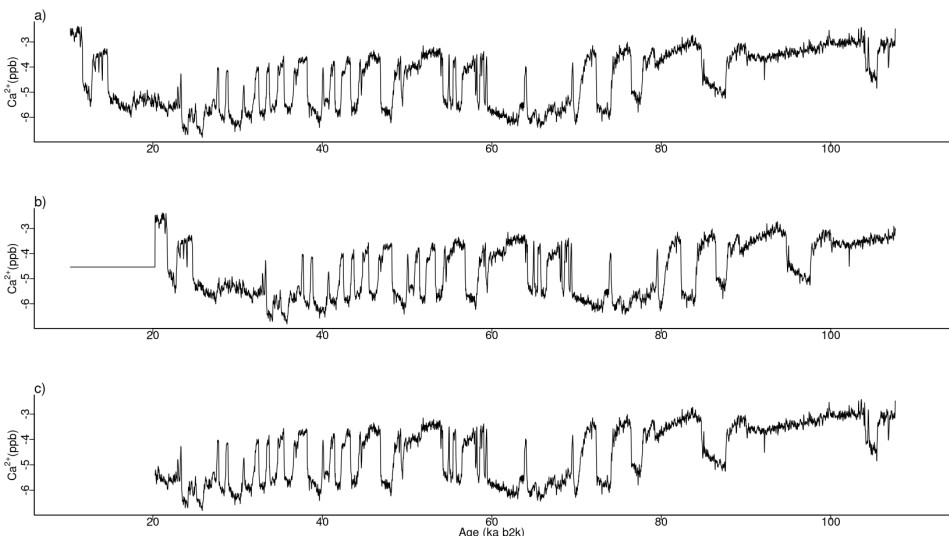

**Figure 13.** Various time series of $Ca^{2+}$ concentration. a) original values, b) shifted version by 10 ka, with the same size, and c) trimmed version by 10 ka, with smaller size. Same age model than in Figure 1.

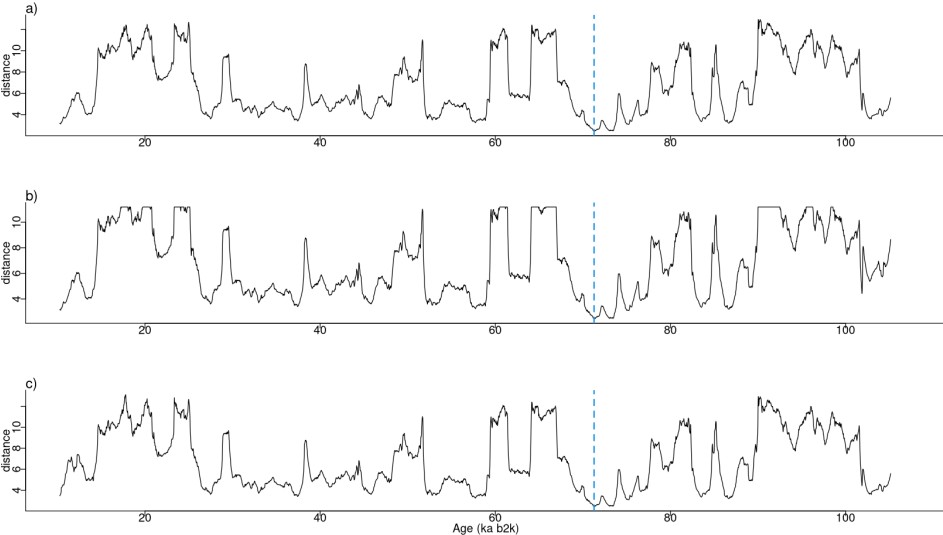

**Figure 14.** Join matrix profile for the $\delta^{18}O$ time series and the $Ca^{2+}$ series in Figure 13 using a window size of 2,500 years. a) original Ca2+ record, b) shifted version, and c) trimmed version. Same age model than in Figure 1.

later. A comparable outcome is observed in the latter case. The profile index indicates that the closest sub-sequence occurs at the index value of the time series corresponding to the correct time. Figure 15 displays the top motif across the $\delta^{18}O$ time series and the different versions of the $Ca^{2+}$ series. It can be seen that the most similar sub-sequence across the two records





corresponds to canonical DO-19. Furthermore, the same results are obtained with shifted and trimmed versions of the $Ca^{2+}$

time series, demonstrating the robustness of the approach.

**Table 5.** Snippet of the matrix profile and profile index for the shifted $Ca^{2+}$ time series. Same convention than for Table 4.

| Sub-sequence ($i$ (time, ka b2k)) | Join matrix profile (distance) | Profile index ($j$ (time, ka b2k)) |
|---|---|---|
| ... | | |
| 3049 (71.22) | 2.50 | 3549 (81.22) |
| 3050 (71.24) | 2.49 | 3550 (81.24) |
| 3051 (71.26) | 2.46 | 3551 (81.26) |
| **3052 (71.28)** | **2.44** | **3552 (81.28)** |
| 3053 (71.30) | 2.45 | 3553 (81.30) |
| 3054 (71.32) | 2.46 | 3554 (81.32) |
| ... | | |

**Table 6.** Snippet of the matrix profile and profile index for the trimmed $Ca^{2+}$ time series. Same convention than for Table 4.

| Sub-sequence ($i$ (time, ka b2k)) | Join matrix profile (distance) | Profile index ($j$ (time, ka b2k)) |
|---|---|---|
| ... | | |
| 3049 (71.22) | 2.50 | 2549 (71.22) |
| 3050 (71.24) | 2.49 | 2550 (71.24) |
| 3051 (71.26) | 2.46 | 2551 (71.26) |
| **3052 (71.28)** | **2.44** | **2552 (71.28)** |
| 3053 (71.30) | 2.45 | 2553 (71.30) |
| 3054 (71.32) | 2.46 | 2554 (71.32) |
| ... | | |

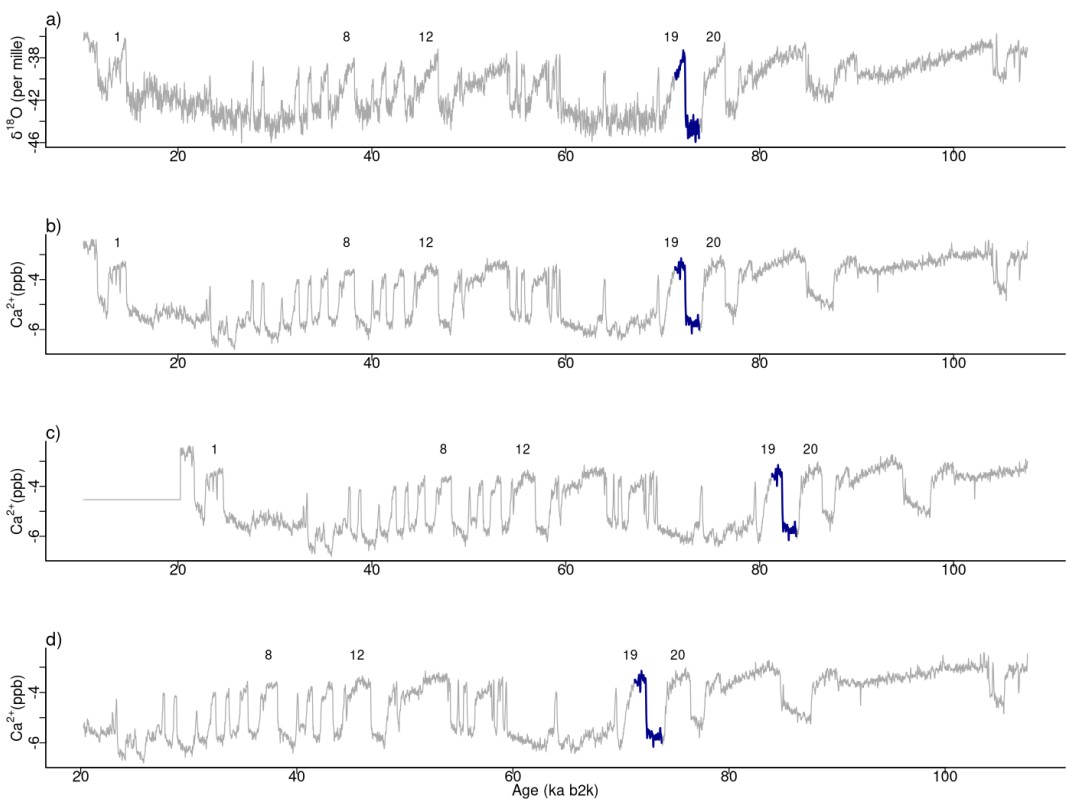

**Figure 15.** Top motif across the $\delta^{18}O$ and the $Ca^{2+}$ time series. a) $\delta^{18}O$ record, b) original $Ca^{2+}$ times series, c) shifted version, and c) trimmed one. Same age model than in Figure 1.


## 6   Conclusions

In this study, the algorithmic matrix profile approach was employed to identify recurring patterns in the well-studied NGRIP palaeoclimate record. The matrix profile is dependent on a single parameter, the sub-sequence length. This is generally set

by by considering the typical duration of the patterns of interest, as there are no stringent criteria for its specification. In this analysis, a window size of 2,500 years was considered, and consistent patterns were obtained for window sizes of 3,000 and 3,500 years, indicating that the results are robust to window sizes within this range.

The objective of the matrix profile approach was not to identify the DO abrupt transitions, or to determine their precise timing. Rather the objective was to characterise the abrupt transitions, in a purely data-driven manner, based on the shape of

the corresponding DO patterns. For the $\delta^{18}$O time series, the transitions corresponding to canonical events DO-19 and DO-20, occurring at around 72 ka and 76 ka, respectively, are identified as the most similar. These are the most prominent top motif pair in the time series, indicating that analogous mechanisms may have been responsible for these abrupt climate transitions. Further transitions corresponding to the events DO-12, DO-8 and DO-1 events are established as neighbouring transitions, with a similar shape characterised by an abrupt transition to warm conditions preceded by approximately stable stadial conditions

and followed by a slow return to cold conditions. The same transitions are identified in the $Ca^{2+}$ time series the same transitions are identified but their ordering differs. The transitions corresponding to events DO-8 and DO-12 being identified as the most similar in the $Ca^{2+}$ record. These events are distinguished by an abrupt decrease in terrestrial dust concentration, followed by a period of stable dust conditions.

The matrix profile method has also been employed to identify the most analogous pattern across the two different $\delta^{18}$O and

$Ca^{2+}$ time series, despite their disparate lengths. Given the high degree of similarity between the two records, the assumption of their simultaneous change, within the 20-year resolution of the records, serves as the basis for the definition of stratigraphic events in Rasmussen et al. (2014). The join matrix profile identifies a coincident top motif, which also corresponds to the DO-19 canonical event. When considering a shifted version of the $Ca^{2+}$ time scale, the join matrix profile is able to identify the same sub-sequence as the most similar pattern across the two time series, although not coincident in time. This allows for

the accurate identification of the time shift that was introduced. A shorter version of the $Ca^{2+}$ time series, also demonstrates the ability of the join matrix profile to correctly identify matching patterns across the $\delta^{18}$O and $Ca^{2+}$ records. This indicates the potential of the join matrix profile as an objective quantitative approach for matching palaeoclimate time series, as an alternative to visual "wiggle-matching". The identification of similarities in events across distinct proxy records can assist the investigation of the key factors affecting the marine ($\delta^{18}$O) and continental (dust) hydrological cycles during the last 130,000

years. This can be tested by Earth System Models.

*Code and data availability.*   All the data and software code used in this work are publicly available at https://rdm.inesctec.pt/dataset/cs-2024-002 (Barbosa et al., 2024)





*Author contributions.* SB: Conceptualization; Formal analysis; Writing—Original Draft Preparation; ES: Writing—Review and Editing DDR: Writing—Review and Editing.

*Competing interests.* The authors declare no competing interests

*Acknowledgements.* This project is TiPES contribution #281: This project has received funding from the European Union's Horizon 2020 research and innovation programme under grant agreement No 820970. Part of this work is supported by Fundação para a Ciência e Tecnologia LA/P/0063/2020.





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
