# Peer review of "Characterisation of Dansgaard-Oeschger events in palaeoclimate time series using the Matrix Profile"

_Nonlinear Processes in Geophysics, 2024_

## Author Comment (AC1)

**Response to reviewers**

We are grateful to both reviewers for their evaluation of our work and for the helpful comments and suggestions. A summary of changes implemented in the revised version is provided below, followed by our point-by-point response to the reviewer comments.

Summary of changes in the revised version

- Added to section 2 (methodology) mention to the computational complexity of the matrix profile algorithms;

- Added to section 4.1 a more detailed description of Table 1;

- Added to section 5 a mention of the utility of applying the matrix profile approach to a single time series and to two time series;

- Added to section 5.1 further clarification on extraction of higher order motifs beyond the top motif, a note on statistical significance of motifs, as well as text on methodological extensions beyond the scope of the present study;

- Added to section 6 (1$^{st}$ paragraph) a note on short-term transitions not being addressed in the study;

Reviewer 1

*In the manuscript "Characterisation of Dansgaard-Oeschger events in palaeoclimate time series using the Matrix Profile" Susana Barbosa, Maria Eduarda Silva, and Denis-Didier Rousseau use the matrix profile method to analyse isotope time series from the NGRIP ice cores. The aim of the study is not to date the various tipping points in the record but rather to demonstrate the potential of a purely data-driven approach in charaterise the abrupt transition. Using the matrix profile approach allows a quantitative identification of typical transition motifs that are associated with the DO transitions the authors focus on. Moreover, the method allows to measure the similarity between transitions and not surprising the transition motifs for DO-19 and DO-20 are identified as the most similar.*

*The manuscript is nicely written and the results are given by meaningful figures and tables. Overall, I enjoyed reading the manuscript. The authors give a good summary of the matrix profile method. This section is detailed enough to understand the method applied and the references given to the original works allow a deep-dive into the method if required. delta-O-18 and Ca2+ records are analysed, first by themselves (finding self-similarity) and then together (focussing on "cross"-similarity/join matrix profile). The investigations follow a logical structure and in combination show the strength of the method studied. The study is completed by testing the limits and dependencies of the method on only parameter the method depends on and on including an artificial time shift between the delta-O-18 and Ca2+ records.*

Thanks for your comments.

*While the manuscript is a good example of how novel time series analysis methods can be applied in a paleoclimate context, research never ends and consequently I missed two main points in the manuscript. Firstly, the strength of the matrix profile method is to find motifs in time series in a computational efficient way. The authors should make clear how the record under consideration with only 4869 data points requires such an advanced method and why traditional methods are computational not tractable.*

We added to the revised version (section 2) information on the computational complexity of the matrix profile algorithm, and its advantage relative to brute force traditional approaches for calculation of euclidean distances between all sub-sequences in a time series. In the case of short time series, such as the one we

consider in this study, the computation of the matrix profile is tractable anyway, it's main advantage lies on the subsequent analysis that can be performed after having calculated the matrix profile, in this case the computation of motifs.

*Second, while the manuscript contains some test demonstrating the stability of the methods and results, strictly speaking there is no consideration given with respect to the statistical significance of the results. While bootstrapping would not make sense in the context of this study, more advanced surrogate data methods, like the Small-shuffle technique (T. Nakamura & M. Small Phys. Rev. E 72, 056216 (2005)), could be used. With such methods one could investigate how destroying for example the short term correlation in the data changes the distance matrix. I understand that such an investigation might well be beyond a simple revision and might by itself be the basis for a future publication.*

We agree with the reviewer, and we modified the revised version (section 5.1) in order to mention explicitly available approaches to evaluate the statistical significance of the results (when considering multiple motifs rather than focusing on the extraction of the top motif, as was the case in our study). We also added to section 5.1 future extensions of this work, including the possibility indicated by the reviewer, and the issue of taking into account measurements and dating uncertainty. These are indeed extensions to be considered in future work rather than in the current revision.

*I also found a small number of typos in the manuscript:*

*l57: taht -> that*

*l181: vales -> values*

*l245: by by -> by*

Thanks for spotting the typos, all were corrected in the revised version.

**Reviewer 2**

*Review of "Characterisation of Dansgaard-Oeschger events in palaeoclimatetime series using the Matrix Profile" by Barbosa et al*

*Barbosa and colleagues apply the pattern-recognition algorithm called the Matrix Profile on NGRIP ice-core measurements of d18O and Ca+ in order to provide an objective ad automatic characterisation of Dansgaard-Oeschger warming events. The algorithm successfully identifies and matches a number of events according to their similarity. The identification of so-called motifs works within one time series as well as across different time series. It is even able to identify corresponding patterns if the compared records are of different length or if the time axis is shifted.*

*While the method seems to work well with the presented data set, the implications and relevance of the study remain unclear. The discussion of the results remains superficial and very much within the space/jargon of the matrix profile. It does not become clear to me what the method can achieve that could not have been achieved by other means. I outline my major comments below and suggest that these comments need to be addressed before the paper can be considered for publication.*

Thanks for your comments, which we took into account in the revised version. In terms of what the method can achieve, we make a simple claim of automatic identification of the most similar sub-sequences in a time series, or across two time series, in a computationally-efficient way. Although computational efficiency is not a clear advantage in this case, as the time series is quite short, the approach is nevertheless quite effective in achieving this objective with minimal assumptions. Alternative means, for example clustering-based

approaches, could probably be applied, but wouldn't enable to achieve the same objective in such a parameter-free and simple framework.

*Major Comments*

*1. Relevance/Novelty and physical interpretation*

*- The objective of the study was to "characterise the abrupt transitions, in a purely data-driven manner, based on the shape of the corresponding DO patterns". I have learned a lot about similarity between different DO-events, but the resulting interpretation does not add any new information to what we already new about DO-events. The only physical characterisation is given as "an abrupt transition to warm conditions preceded by approximately stable stadial conditions and followed by a slow return to cold conditions" for d18O and "distinguished by an abrupt decrease in terrestrial dust concentration, followed by a period of stable dust conditions." for Ca+. Both of these statements seem established knowledge and could have been obtained almost purely by eye. I believe the relevance of the study could be greatly improved by giving more physical context and interpretation to the obtained results. I list some example questions below that could help in giving this context:*

We agree with the reviewer that the paper is mainly focused on the identification of similar DO events, and that the results "*could have been obtained almost purely by ey*e". Indeed the key here is the qualifying "almost". Although the human eye is quite effective in identifying patterns, it is also very much influenced by biased perceptions, leading to uncertainty and even erroneous conclusions in human-perceived similarity. Having an objective way of identifying which events are more similar, and having the results confirmed by visual inspection, is not a limitation in itself, but a further convincing evidence of the utility of the method.

*(1) What is the actual advantage of the method with respect to previous methods? The authors mention that it can be a powerful alternative to "wiggle matching" (see also major comment 2). I can see this, but most of the performed analysis was done within one time series, what is the advantage here?*

The advantage relative to the previous approaches mentioned in the introduction, particularly in the analysis of a single time series, is the ability to answer the question of which DO events are most similar to each other, in terms of their shape. Previous approaches do not pose this question, so the methods are not comparable, as they are not addressing the same problem. The main advantage of applying this method to a single time series is the ability to focus on which parts of the time series are more similar to each other - note that DO events are not even directly considered, only the sub-sequence length is selected (sub-sequences of that length which are more similar happen to correspond to DO events).

*2) What do I gain from knowing which events are most similar to each other? Can the information help with stacking or with classification? Here, the authors give a small hint in their conclusion, stating that the identified similarity could be an indication of similar underlying mechanisms. Can this claim be backed up/elaborated upon? And what about the other (shorter) unidentified transitions?*

Again, DO events are not explicitly analysed, the method just "sees" sub-sequences of the same length across the time series. In the present study the extracted motifs and neighbours correspond to DO events, indicating that the recurring patterns in the time series, for sub-sequences of the length considered in the study, do coincide with classical DO events. As we have hinted, it seems likely that two events that are very similar, in terms of their shape, and therefore warming/cooling pattern for $\delta^{18}O$ or dustiest/less dusty for $Ca^{2+}$, would be associated to similar physical mechanisms, but that is beyond the scope of this work, especially when comparing the two different proxies, as our study is mainly focused on the methodological aspects for ascertaining the similarity.

Indeed we were not able, with the work flow used in the paper, to identify shorter transitions, but that doesn't mean that those shorter transitions are unimportant or not physically relevant, just that short-length motifs can be harder to identify in terms of recurring patterns. A note on this was added in the revised version (1st paragraph of section 6).

*(3) What is the physical interpretation/relevance of an event being a top motif or a neighbouring motif?*

The top motif identifies the two most similar events, while neighbour motifs correspond to events that while not being the most similar, are still quite similar to those two events.

Please note that motifs in themselves are not necessarily physically interpretable, although they often are. This is somewhat similar to the situation in a totally different method, PCA (Principal Component Analysis), or EOFs, as commonly called in meteorology. The method enables the extraction of patterns of maximum variance, so it yields statistical modes, which often are physically-interpretable patterns (e.g. ENSO), but not necessarily so, that is, physically interpretable patterns may not correspond to maximum variance patterns, and vice-versa. Lacking physical interpretability doesn't mean that there is some problem with the EOF analysis or the data, just that the dominant characteristics of the data are not associated with a maximum variance pattern (the most obvious example is the case of trends).

*(4) What does it imply that the matrix profile identifies different top motifs in d18O and Ca+?*

For the $\delta^{18}O$ record the top motif (most similar events) corresponds to events DO-19 and DO-20, while for the $Ca^{2+}$ record the DO-8 and DO-12 events are the most similar. Although the ordering of the motifs is different, qualitatively the results are very similar, as the same motifs are extracted. Because the two records have similar temporal variability but are not exactly identical, quantitatively the sequences that are the most similar in the $\delta^{18}O$ record are still very similar, though not the most similar, in the $Ca^2$ record. It implies that despite the apparent same temporal variability, the two proxy records are associated with distinct climate factors, and the two time series are not identical. The method is able to pick-up those quantitative differences, while providing consistent qualitative information.

*I think possible limitations of the method need to be discussed in more detail, especially when it comes to the joint matrix profile. The test data set used here are two records on the same age model from a similar location. These seem very favourable conditions. What if the records have different temporal resolution as is often the case? How would the method be used to synchronise two records if they have different age models? And as I understand it, the method can only be used to synchronise two time series if the manifestation of the events follows a similar pattern. What if there is a phase shift or if the pattern is not as pronounced as in the Greenland ice cores?*

The method is not a silver-bullet for synchronization of paleoclimate records, and certainly has limitations. One of the most obvious limitations is that the uncertainty information available for the proxy records is not taken into account. That would be a relevant future extension of the methodology, and we included a note on this in the revised version, at the end of section 5.1.

Our approach doesn't use any age model information, relying entirely on the shape of the patterns. That is both a blessing and a curse. Not using an age model, as the method doesn't need that information, is an advantage – as we have demonstrated in section 5.2 – but looking only to the shape of the signal might be a strong disadvantage if the shape of the signal does not reflect accurately the underlying process. The matrix profile approach doesn't solve the many subtle and challenging aspects intrinsic and specific to

palaeoclimate time series, but is an useful automated alternative to visual inspection that can be used to gain insight on paleoclimate records.

*Also, as I understand from the discussion, the window size does not seem to have a big effect on the identification of main motifs in the range of 2500-3500 years. But what about the smaller window sizes and shorter DO-events? Currently only long events are captured and shorter events (e.g. events 3-7, 9 and 10) are probably ignored because of the chosen window length. Where is the lower bound for meaningful results?*

We have added to the revised version ($1^{st}$ paragraph of section 6) explicit mention to the issue of identification of shorter DO events.

*3. Clarity -----------------------------------------------*

*In parts, it is difficult to follow the method description and interpretation, especially because the word "close" is used for being similar and for being close in time. I suggest to use "similarity" instead of "distance" thoughout the text to avoid confusion between close/distant in time and similar/dissimilar according to the Euclidian distance. E.g. line 53-54 would become: "The matrix profile [...] that stores the Euclidian distance [...] to its most similar sub-sequence. The similarity is measured using the Euclidian distance [...]."*

We implemented this suggestion in the revised version.

*Minor Comments*

*paragraph starting in l52 - it might be good to specifiy early on how the different sub-sequences are defined. That information only comes in l72 but would be good to have earlier.*

Updated in the revised version.

*l73-76 - in my understanding, applying the joint matrix profile to two time series is where the real power and benefit of this method lies. I think this should be highlighted more!*

We agree with the perceived potential of the join matrix profile, but the power and benefit of the method is not at all restricted to two time series, abundant literature exists showing the benefit of the method in the case of a single time series. It might be less obvious in the case of short time series, such as the one analysed here, but finding motifs in a single long and high-frequency time series is a very common need in data mining contexts, for which the matrix profile, of a single time series, is considered to be an extremely useful tool. Text clarifying this aspect was added to section 5.

*l94-100 - this seems to be a more specific version/repetition of the method section. Think about incorporating it there instead?*

We understand the point, and considered indeed whether to move it to the method section. Still we would prefer to keep it here, to improve clarity and the presentation of the results, making easier for the reader to follow what are the *i* and *j* mentioned in the tables and what are the numbers presented in the tables.

*l103-112 - is the matrix profile symmetric? If yes, should there not be matrix profile of the two closest sub-sequences not be identical, resulting in two global minima? How do you decide which one to pick as the top motif?*

No, the matrix profile doesn't need to have two global minima. The top motif is the one corresponding to the minimum value of the matrix profile. Suppose we have a time series with 5 data points (n=5), and we consider sub-sequences of 2 data points (m=2). Sliding a window of size m=2 through this time series, yields 4 (=n-m+1) sub-sequences of length 2, specifically sub-sequence #1 (let's call it S1) including datapoints 1 and 2, sub-sequence #2 (S2) including data points 2 and 3, sub-sequence #3 (S3) with data points 3 and 4 and

sub-sequence #4 (S4) including data points 4 and 5. The matrix profile is a vector of the same length as the number of sub-sequences (so in this case of length 4). The first element of the matrix profile vector is the minimum value of a set of 3 distances (between S1 & S2, S1 & S3, and S1 & S4). Let's suppose that the smallest distance is between S1 and S4. Likewise, the 4th element of the matrix profile vector is the minimum value of a set of 3 distances (between S4 & S1, S4 & S2, and S4 & S3). In these two cases (of the 1st and 4th values of the matrix profile) the distance value is the same between sub-sequences S1&S4 and S4&S1, but note that the minima are not necessarily the same (and in general they aren't), that is

minimum (S1 & S2, S1 & S3, and S1 & S4) ≠ minimum (S4 & S1, S4 & S2, and S4 & S3)

*l114-124 - I find this very confusing. Please explain more carefully. It may also be good to have d18O, the matrix profile and the matrix profile index all in one figure. The distances in Table 2 are not the same distances as shown in Fig1, right? So to identify neighbouring motifs you have to perform additional calculations to get the distance between the top motif and all other possible sub-sequences? What is the difference of neighbouring motifs to 1st and 2nd order motifs mentioned in the method section?*

The matrix profile method is conceptually very simple, but the outputs of the algorithm can indeed be hard to follow, and that's the reason why we made an effort to present the results in a detailed way, keeping figures not showing the same thing separate, and presenting snippets of the algorithmic output, to facilitate understanding of the results.

We made an effort in the revised version to explain more clearly the results, adding additional explanations to section 4.1 and to the description of Table 1.

The distances in table 2 (middle column) are the same as in Figure 2, so around 71 ka the value of the matrix profile is about 2.3, as can be seen in Figure 2. The (global) minimum of the matrix profile (indicated by the vertical dashed line in Figure 2, and the bold numbers in Table 1) indicates the top motif. So 2.34 is the smallest distance between the sub-sequence starting at datapoint # 3045 (71.14 ka) and another sub-sequence, which is given in the profile index, so it is the sub-sequence starting at datapoint #3251 (75.26 ka). Note that no additional calculations are needed – actually the the matrix profile algorithm returns exactly Table 2, but of course we have shown only a portion of that output.

Returning to the previous toy example of a time series with n=5 datapoints and sub-sequences of length m=2, assuming that the smallest distance to the 1st sub-sequence is between S1 and S4, the matrix profile algorithm would output a table having as 1st row:

| sub-sequence | distance of most similar sub-sequences (matrix profile) | Sub-sequence most-similar to the sub-sequence in the 1st column (profile index) |
|---|---|---|
| 1 (S1) | Min (S1&S2, S1&S3, S1&S4) | 4 (S4) |

The difference of neighbouring motifs to 1st and 2nd order motifs is indeed dependent on the adopted strategy for extraction of motifs, and that's why we made a detailed analysis on the impact of that criteria (in particular the specification of the value of the radius R) on our results. While the matrix profile calculation is only dependent on the sub-sequence length (and on the adopted metric to measure similarity, in this case normalised euclidean distance) and the top motif is objectively obtained from the global minimum value of the matrix profile, subsequent neighbours/motifs depend on the adopted choice of the radius parameter R. In this study we explicitly restricted the analysis to the top motif, and its neighbour motifs, using the criteria which we discuss in detail in the paper. In general the extraction of motifs has to take into account the purpose of the analysis and is very much problem dependent, in terms of what are the patterns of interest in a time series (which may differ, even for the same time series, depending on the goal of the analysis).

We added this clarification on extraction of subsequent higher order motifs to the discussion in the revised version (section 5.1).

*l190-207 - same as comment on l114-124: I am again confused. Is the approach for selecting the neighbouring motifs in Figure 12 the same as in Table 2? First you describe in length one way of using R and even discuss you results in the context. But then you say that you are actually using another approach. Also a lot of these two paragraphs seems to repeat itself. Please rewrite more clearly!*

Table 2 presents the motifs and neighbours that are obtained based on the value of the radius parameter R, and we show the results that are obtained using two different vales of R, R=2 and R=3. Figure 12 shows the shape of these same motifs. The radius parameter R is conventionally used in the data mining /computing science literature. In the end we do apply another approach in our study, but we presented before the results that are obtained with the conventional strategy, for clarity. It would be probably simpler to just set a value of R – or start right away with our approach and define the motifs from there – but we wanted to make the analysis as transparent and reproducible as possible, and therefore provided these details for fully documenting our analysis. Algorithmic/purely data-driven approaches easily become difficult to reproduce if not enough detail is provided on the assumptions and criteria used, and we tried to make those as clear as possible.

*Some Editorial Comments*

*l17 - word missing: serves \*as\* an indirect proxy*

Done

*l23-25 - hard to read because of too many parentheses, please rephrase*

Done

*l57 - typos: That \*location\* is stored in the profile index. [...] vector \*that\* stores [...].*

Done

*l74 and many other places - should it be \*joint\* matrix/motif instead of "join matrix/motif" ?*

We agree that joint matrix profile sounds better, but the terminology more commonly used in the technical literature is join (or self-join, in the case of a single time series) and we opted to follow the literature and the methodological references provided in the paper

*l98 - the \*smallest\* value*

Done

*l134 - remove "Thus,"*

Done

*l144 - what do you mean by "on the plot"?*

Modified to convey the meaning of normalised values, as presented in the plot (Figure 7)

*l205 - \*than\* that instead of "that that"*

Done